# Benthic Macroinvertebrate Communities in Wadeable Rivers and Streams of Lao PDR as a Useful Tool for Biomonitoring Water Quality: A Multimetric Index Approach

Jutamas Sripanya [1] , Chanda Vongsombath [2] , Viengkhone Vannachak [3] , Kaewpawika Rattanachan [4] , Chutima Hanjavanit [1] , Wuttipong Mahakham [1,5] and Narumon Sangpradub [1,*]

1   Department of Biology, Faculty of Science, Khon Kaen University, Khon Kaen 40002, Thailand
2   Department of Environmental Science, Faculty of Environmental Sciences, National University of Laos, Vientiane 7322, Laos
3   Department of Biology, Faculty of Science, National University of Laos, Vientiane 7322, Laos
4   Department of National Parks, Wildlife and Plant Conservation, Bangkok 10900, Thailand
5   Applied Taxonomic Research Center, Faculty of Science, Khon Kaen University, Khon Kaen 40002, Thailand
*   Correspondence: narumon@kku.ac.th

**Abstract:** Lao PDR, a landlocked country in the lower Mekong River basin of Southeast Asia, has been considered a global biodiversity hotspot with a high level of biological endemism. In recent years, urban development and industrialization have affected the water quality of freshwater ecosystems in Lao PDR. However, the assessment of water quality in the country is primarily focused on a physicochemical method, while the application of a multimetric index (MMI) approach using benthic macroinvertebrates for biomonitoring in rivers and streams has not been established. MMI, based on benthic macroinvertebrates, is a biomonitoring tool that considers the effects of multiple anthropogenic impacts on benthic macroinvertebrate metrics associated with their biological attributes (e.g., taxa richness, composition, pollution tolerance, habits, and functional feeding) and aggregates individual metrics into a single value for assessing the water quality and health conditions of aquatic ecosystems. Here, we developed an MMI based on macroinvertebrate communities collected during 2016–2018 from 10 localities of streams and wadeable rivers in Lao PDR. Of the 54 potential metrics tested, 35 candidate macroinvertebrate metrics representing richness, composition, trophic structure, habit, and tolerance to pollution were selected, while 19 metrics were excluded. Of the 35-candidate metrics, a total of 11 core metrics (Total taxa, EPT taxa, Ephemeroptera taxa, %Diptera, %Plecoptera, %Tolerant, Beck's biotic index, %Intolerant, Filterers taxa, %Sprawlers, and %Burrowers) were finally selected for the development of MMI based on their sensitivity, redundancy, and easy-to-apply tool for the biomonitoring program. These metrics can be used to distinguish the reference (seven sites) from stressed conditions (seven sites). In addition, the final MMI scores classified 40 sampling sites into four classes of water quality, including excellent (25%), good (10%), fair (60%), and poor (5%), which the conventional physicochemical method could not clearly distinguish. The Lao MMI developed in this study is an effective tool for evaluating the water conditions of sites affected by human activities, particularly agricultural areas, and, thus, is appropriate for use in future studies for assessing the ecological conditions of rivers and streams in the Mekong region.

**Keywords:** bioassessment; benthos; Mekong region; multimetric index; water quality

## 1. Introduction

The Lao People's Democratic Republic, or Lao PDR, is a landlocked country located in mainland Southeast Asia. Boarded by China, Vietnam, Cambodia, Thailand, and Myanmar, the country encompasses an area of 236,800 square kilometers and has a population of approximately 6.9 million [1]. As a global biodiversity hotspot with high levels of biological endemism, Lao PDR boasts diverse aquatic ecosystems, including lakes, ponds, rivers, and

streams, which are home to a wide range of fish, amphibians, mollusk, and small aquatic invertebrate species, as well as aquatic macrophytes and algae [2–4]. These freshwater ecosystems play a vital role in supporting food and livelihood security for Lao PDR's fisheries [5]. Thus, the aquatic ecosystems of Lao PDR are of great scientific interest and economic importance. However, in recent decades, Lao PDR's aquatic ecosystems have been severely threatened by rapid population growth, urban development, industrialization, land use changes, dam construction, mining excavation, and the removal of riparian vegetation [6]. According to estimates from the last decade, about 35% of liquid effluent discharged into inland surface waters has been treated, and the quantity is unknown [7]. Furthermore, high concentrations of nutrients ($NO_3$ and P), nitrogenous matter, and total suspended solids (TSS) have been observed in the Mekong River at Vientiane City in Lao PDR [8]. Additionally, the lower Mekong countries, including Lao PDR, are increasingly concerned about the environmental and agricultural damage caused by hydropower development and dam construction [6]. Taken together, these anthropogenic activities have resulted in the deterioration of water resources, which, in turn, has had a negative impact on the aquatic environment and biodiversity in Lao PDR.

To achieve the sustainable development and management of water resources, it is crucial to understand the relationship between human activity and the health of aquatic ecosystems, as well as to develop reliable tools for monitoring water quality. Historically, water quality has been primarily evaluated using physicochemical methods, which analyze various physical and chemical parameters of water against the established standards [9,10]. However, these methods are insufficient in assessing the consequences of certain human disturbances, as they only reflect the water quality at the time and place of sampling and do not directly measure the biological response to pollution [11,12]. Therefore, physicochemical methods cannot detect biogeochemical changes that occur in aquatic ecosystems over an extended period of time [13]. On the other hand, biomonitoring methods can reflect the consequences of changes in water quality and the biological response to human disturbance and pollution, both in the past and present [12,14–18]. Therefore, biomonitoring can provide a more accurate assessment of the true health of aquatic ecosystems, making it a valuable tool when used in conjunction with physicochemical approaches.

Benthic macroinvertebrates are a commonly used group of animals for biomonitoring in freshwater ecosystems [14,19]. These organisms are taxonomically and functionally diverse, with each group having specific environmental needs and ecological preferences. As a result, they show varying responses to changes in the ecological conditions of their habitat and water quality [17]. Environmental factors such as dissolved oxygen (DO), biological oxygen demand (BOD), nitrate nitrogen content, phosphorus compounds, water velocity, and depth, as well as the quality and quantity of available habitats, have been found to directly and indirectly, affect the diversity, composition, and distribution of benthic macroinvertebrates [15,20–24]. For example, in freshwater habitats with low DO concentrations, tolerant taxa such as Mollusca and Oligochaeta are often prominent [25]. Habitat heterogeneity in streams and rivers can also influence the functional composition and diversity of benthic macroinvertebrates. Research has shown that high levels of habitat heterogeneity promote faunal diversity, particularly among benthic macroinvertebrates [15]. Substrate and sediment grain size also play an important role in regulating the composition, distribution, diversity, and geographic preference of benthic macroinvertebrates. A small sediment grain size in sandy habitats affects the spatial distribution and density of benthic macroinvertebrate taxa [26,27]. Additionally, the decrease in the percentage of pebbles can lead to low habitat diversity for benthic fauna, resulting in a decline in the richness and density of benthic macroinvertebrates [28]. Therefore, changes in substrate conditions resulting from urbanization increased anthropogenic and agricultural activity, and the reduction in riparian vegetation can significantly threaten benthic macroinvertebrates and other aquatic organisms in streams and rivers [29]. As such, benthic macroinvertebrates can be used as indicators of various types of anthropogenic disturbance [30,31].

The use of biological indices for biomonitoring the water quality in freshwater ecosystems has been established for a significant amount of time [11,13–17]. However, there are limitations when using a single index for water quality monitoring [32]. While evaluation based on species occurrence can detect changes in water quality and pollution, these indices are relatively subjective and require knowledge of ecological life strategies and intensive sampling. As a result, a single index is often not able to accurately reflect the overall status of an aquatic ecosystem under various anthropogenic activities [17]. To overcome these limitations, a more integrated approach known as the multimetric index (MMI) has been introduced. This approach has gained increasing attention worldwide for its ability to integrate different environmental drivers that are impacted by human and natural activities over an extended period of time [11,12,14–19,30,32–39]. The MMI approach based on benthic macroinvertebrates has become a popular tool for biomonitoring programs in the United States [15,30,34] and European countries [32,35,40], as well as for monitoring the ecological health of rivers and streams in Asian countries such as China, Korea, Malaysia, Thailand, and Vietnam [17,18,37,41–44]. The MMI approach is flexible and can easily be adapted by adding or removing biological metrics or refining the index threshold values. However, it is important to exercise caution when using MMIs in different ecoregions or environmental gradients in specific geographical regions, which may have different reference conditions, anthropogenic pressures, and regional species assemblages [19,37].

In Lao PDR, water quality assessment is primarily focused on physicochemical and microbial analysis [45,46], while the use of biomonitoring is still relatively new [47]. Additionally, there is a lack of research on the application of the MMI approach using benthic macroinvertebrates for the biomonitoring of rivers and streams in Lao PDR. Furthermore, there is limited knowledge of the ecology and biodiversity of rivers and streams in this country, making the Mekong ecoregion in Lao PDR particularly important for assessing its current aquatic ecological conditions. The main objective of this study is to develop an MMI for biomonitoring wadeable rivers and streams in Lao PDR. To the best of our knowledge, this study is the first attempt to create a "Lao MMI": an easy-to-apply and cost-effective tool for monitoring and evaluating the ecological condition of rivers and streams in Lao PDR.

## 2. Materials and Methods

### 2.1. Study Site

In the present study, 10 localities situated in three main ecoregions, the Lower Lancang, Khorat Plateau, and Kratie-Stung Treng, were used as sampling areas (Figure 1, Table 1). Reference sites were chosen based on the minimally disturbed areas, while the stressed sites were primarily assessed based on the presence of anthropogenic activities close to the studied areas. Samplings were conducted four times during the cool (December 2016 and November 2017) and hot seasons (April 2017 and 2018).

### 2.2. Measurement of Environmental Variables

Habitat assessment was taken at each site following the method of the USEPA [15]. Ten parameters of physical habitats were assessed, including the epifaunal substrate/available cover, embeddedness, velocity/depth regime, sediment deposition, channel flow status, channel alteration, frequency of riffles (or bends), bank stability, vegetative protection, riparian vegetative zone width. Total habitat scores (THS) were calculated and ranged from 0−200 points. THS values <130 points were considered as the stressed physical habitat condition [48].

Physico-chemical characteristics of water (15 parameters) were measured along with the collection of benthic macroinvertebrates at each of the sampling sites. Air and water temperatures (°C) were measured using a thermometer, while dissolved oxygen (DO, mg/L) was determined using a dissolved oxygen meter (YSI model 550A, Yellow Spring Instrument Co., Inc., Yellow Spring, MP, USA). Electrical conductivity (EC, μS/cm), total dissolved solids (TDS, mg/L), and pH were collected using a portable multi-probe meter

(Hanna model HI 98129, Hanna Instruments, Woonsocket, Rhode Island, USA). In addition, 2 L water samples from each site were collected in sterilized plastic bottles and transported in an ice box (approximately 20 °C) to the laboratory for the standard measurement of orthophosphate ($PO_4^{3-}$, mg/L), nitrate nitrogen ($NO_3$-N), and chlorophyll a (μg/L) using previous methods [18,49]. For the five-day biochemical oxygen demand ($BOD_5$) analysis, water samples were stored in dark-colored glass bottles at room temperature. The initial $BOD_5$ concentration was measured with a dissolved oxygen meter (YSI model 550A). After an incubation period of 5 days at 20 °C in the dark, the water sample was checked again for DO content. The $BOD_5$ was calculated from the difference in oxygen content between the start and end of the measurement [49]. Suspended solids (SS, mg/L) were determined using the photometric method [50]. Turbidity (NTU) was measured with a turbidimeter (Hach model 2100N, HACH Company, Loveland, CO, USA). Moreover, water channel width (m) and depth (cm) were measured using tape and a steel rod, respectively, while water velocity (m/s) was recorded using a pygmy water current meter (Genuine Gurley® current meter model D625 digital pygmy meter with Model 1100 Flow Velocity Indicator, Gurley Precision Instruments, New York, NY, USA).

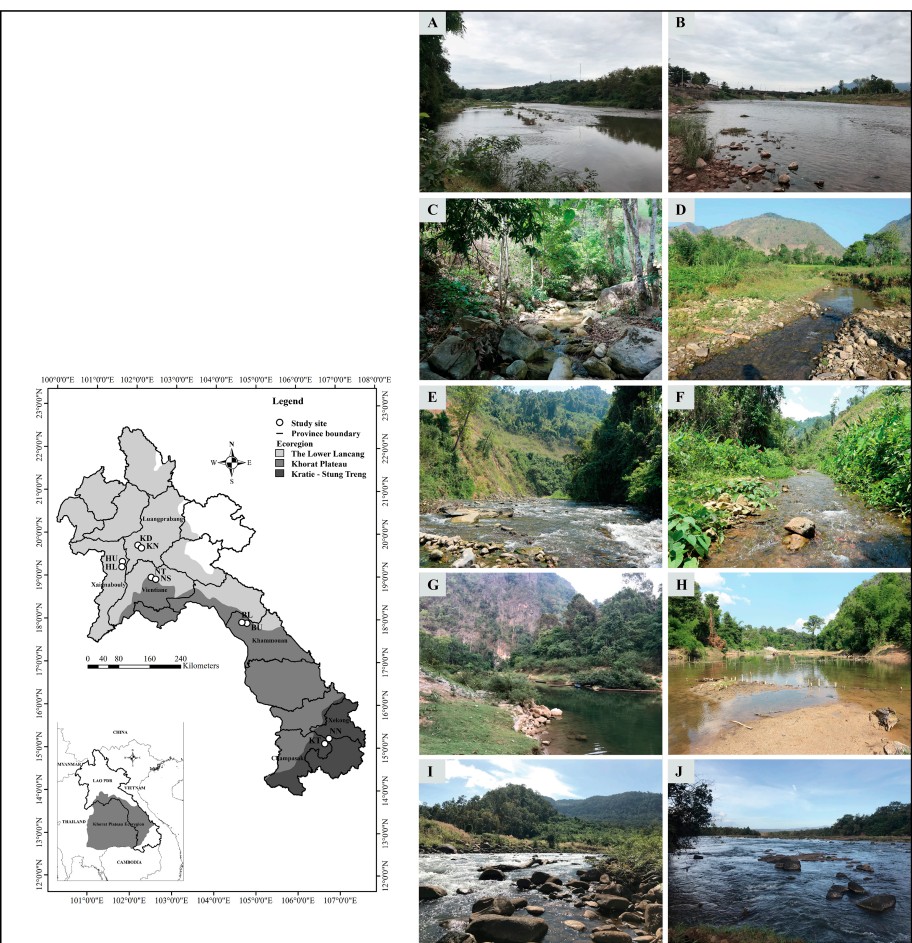

**Figure 1.** Map showing the distribution of the 10 study localities (wadeable rivers and streams) in Lao PDR. Photographs of 10 study localities. (**A**) Nam Houng upstream (HU), (**B**) Nam Houng downstream (HL), (**C**) Nam Khod (KD), (**D**) Nam Khan (KN), (**E**) Nam Song (NS), (**F**) Nam Thang (NT), (**G**) Nam Hinboun upstream (BU), (**H**) Nam Hinboun downstream (BL), (**I**) Xe Katam (KT), (**J**) Xe Namnoy (NN).

**Table 1.** General characteristics and geographical position of 10 localities of wadeable rivers and streams conduced in this study in Lao PDR.

| Ecoregion | Study Localities | Type of Lotic Ecosystems | Province | Coordinates | Altitude (masl) | Activity | Substrate Types (%) |
|---|---|---|---|---|---|---|---|
| The Lower Lancang | 1. Nam Houng upstream (HU) | Wadeable river | Xaignabouly | 19°18′30.68″ N 101°43′24.94″ E | 294 | Forest areas, small agriculture patches | Cobble (60%), pebble (10%), gravel (20%), sand (10%) |
| | 2. Nam Houng downstream (HL) | Wadeable river | Xaignabouly | 19°15′38.89″ N 101°42′46.01″ E | 283 | Residential areas, agriculture areas | Cobble (70%), gravel (10%), sand (15%), detritus (5%) |
| | 3. Nam Khod (KD) | Stream | Luangprabang | 19°44′14.12″ N 102°8′52.63″ E | 375 | Forest areas, small agriculture patches | Boulder (20%), cobble (30%), pebble (15%), gravel (20%), sand (5%), detritus (10%) |
| | 4. Nam Khan (NK) | Stream | Luangprabang | 19°43′55.58″ N 102°9′24.31″ E | 332 | Agriculture areas, intensive erosion | Cobble (50%), detritus (10%), muck-mud (40%) |
| Khorat Plateau | 5. Nam Song (NS) | Wadeable river | Vientiane | 19°6′13.07″ N 102°30′3.16″ E | 315 | Forest areas, small agriculture patches | Boulder (10%), cobble (65%), pebble (15%), gravel (5%), sand (5%) |
| | 6. Nam Thang (NT) | Stream | Vientiane | 19°6′15.12″ N 102°29′54.98″ E | 324 | Forest areas, small agriculture patches | Cobble (60%), pebble (15%), gravel (10%), sand (10%), detritus (5%) |
| | 7. Nam Hinboun upstream (BU) | Wadeable river | Khammouan | 17°57′19.47″ N 104°45′28.73″ E | 164 | Preserve areas | Cobble (15%), pebble (5%), gravel (60%), sand (20%) |
| | 8. Nam Hinboun downstream (BL) | Wadeable river | Khammouan | 17°57′33.58″ N 104°43′32.96″ E | 156 | Agriculture area | Gravel (20%), sand (40%), muck-mud (40%) |
| Kratie-Stung Treng | 9. Xe Katam (KT) | Wadeable river | Champasak | 15°7′48.93″ N 106°40′09.92″ E | 257 | Forest areas, small agriculture patches | Boulder (60%), cobble (30%), gravel (5%), sand (5%) |
| | 10. Xe Namnoy (NN) | Wadeable river | Xekong | 15°13′37.99″ N 106°44′45.30″ E | 132 | Forest areas, small agriculture patches | Boulder (50%), cobble (30%), pebble (10%), gravel (5%), sand (5%) |

### 2.3. Benthic Macroinvertebrate Sampling

Benthic macroinvertebrates were sampled from each site using the multihabitat approach by a D-frame net ($30 \times 30$ cm$^2$, 450 µm mesh size). A total of 20 kicks were collected from all habitats within a 100 m stretch and combined into a single sample. Animal samples were immediately fixed in 95% ethanol for observation in the laboratory. Then, they were sorted, subsampled, and counted for $300 \pm 60$ individuals [18]. Benthic macroinvertebrates were identified to the finest possible taxonomic level (usually family or genus) by compound and stereo microscope using the available identification keys [51,52]. All individuals were counted following identification. In addition, they were classified into a habit, functional feeding groups, and tolerance values according to previous reports [52–59]. Some taxa with unknown tolerance values were assigned based on their distribution in land use.

### 2.4. Site Classification

Based on the preliminary investigation, most of the study localities had quite a similar sediment composition except for Nam Houng downstream (HL), Nam Khan (NK), and Nam Hinboun downstream (BL), which demonstrated sites of human disturbances with less gradient substrate types. While Nam Houng upstream (HU), Nam Khod (KD), Nam Song (NS), Nam Thang (NT), Nam Hinboun upstream (BU), Xe Katam (KT), and Xe Namnoy (NN) were sites with fewer human stressors and were defined as having characteristics of dense riparian vegetation areas with high gradient substrate types (Figure 1, Table 1).

The reference sites used for developing a multimetric index were selected in the areas that were as minimally impaired as possible based on the site classification criteria [37,44]. Sample sites were classified into reference, intermediate, and stressed sites based on the following 14 environmental parameters: dissolved oxygen (DO), pH, electrical conductivity (EC), nitrate nitrogen ($NO_3$-N), percent of land use (%land use), epifaunal substrate score (EpifauSub score), velocity score, sediment deposition score (SedDep score), bank stability score (BankSta score), vegetative protection score (VegPro score), percent of total habitat score (%THS), dam present, point source present, and the percentage of benthic macroinvertebrate taxa (%BMC taxa) [34,37,44,48,60]. A sampling site was classified as a reference site when all 14 criteria reached a standard or good level; sampling sites were classified as stressed sites if only 1 criterion failed with poor water quality or a high degree of development pressure [44]. The study site was designated as an intermediate site, as it did not fulfill all 14 criteria used to define a reference site and also did not meet any of the 14 criteria used to identify a stressed site.

### 2.5. Data Analysis

#### 2.5.1. Habitat Score, Environmental Variables and Benthic Macroinvertebrate Composition

The mean $\pm$ SD of habitat scores, environmental parameters, and benthic macroinvertebrate communities were calculated for reference, intermediate, and stressed site groups. According to data distribution, one-way ANOVA or the Kruskal–Wallis test was used to determine significant differences in the total habitat score, environmental parameters, and benthic macroinvertebrate communities among the three groups (reference, intermediate, and stressed sites). In addition, if two groups were tested, the Man-Whitney U test was used to determine their statistical difference. All statistical analyses were performed using the IBM SPSS Statistics version 23 software [61]. Canonical correspondence analysis (CCA) was used to find relationships between environmental variables and benthic macroinvertebrate communities using PC-ORD software (version 5) [62].

#### 2.5.2. MMI Development

Similarities in the benthic macroinvertebrate faunal composition between ecoregions were calculated with the ANOSIM by Past (Paleontological Statistics) version 4.03 [63]. An R-value close to one indicated a high difference, and close to zero indicated a low difference in the faunal composition between ecoregions. Independent sample T-test analysis was used to test significant differences in the number of benthic macroinvertebrate

taxa between seasons and years. The benthic macroinvertebrate data from each sampling site was imported into the Ecological Data Application System (EDAS) version 3.3 for data management and the calculation of metrics [64].

The development of the macroinvertebrate-based MMI for Lao PDR was adapted from the procedure guided by the USEPA [15]. Fifty-four candidate metrics covering five categories of benthic macroinvertebrates, i.e., richness, composition, tolerance/intolerance, functional feeding group, and habit, were screened and examined according to the previous report [44]. The suitability of metrics for discriminating the reference and stressed conditions was assessed according to previous reports [37,44]. The comparison of the metric value between reference and stressed sites was performed with the aim of selecting the most appropriate metrics for Lao streams and wadeable rivers. The selection of potential metrics from the candidate metrics was based on two applications: sensitivity tests and redundancy tests [19,37].

In the first step of the sensitivity test, the capacity of each metric to differentiate the minimally disturbed sites (reference sites) from the most disturbed sites (stressed sites) was tested and visualized by box and whisker plot analysis. The metrics which showed a clear discriminatory power between the reference and stressed sites, i.e., no overlap of interquartile ranges, were selected [37]. The discriminatory power score of each metric was assessed according to the degree of overlap for medians and interquartile ranges [34]. Metrics with discriminatory power score 3 were mainly selected as potential metrics. If discriminatory power 2 was selected, a high (>70%) discrimination efficiency (DE) was also considered. DE was calculated as the percentage of stressed sites that scored below the 25th percentile of the reference sites for decreasing metrics with disturbance and above the 75th percentile for increasing metrics with disturbance [19,65]. Based on the discriminatory power or sensitivity score, 35 metrics were retained. These metrics were then selected based on their DE value [65].

In the second step, redundancy between the selected metrics was tested and evaluated using Pearson's correlation analysis [37]. Pairs of metrics with $r > 0.85$ were considered redundant [17,41]. Redundant metrics were then removed, and at least one metric per category of benthic macroinvertebrates was retained [66,67]. Apart from the two testing steps, i.e., sensitivity and redundancy tests, ecological importance and the wider applicability of each metric were also used as criteria for selecting redundant metrics into the final core metrics. Spearman's correlation was used to test a correlation among the core metrics, environmental variables, and grain sizes before MMI integration.

The final MMI was constructed using an integration of selected core metrics representing five categories of benthic macroinvertebrates, as mentioned above. Prior to integration, selected metrics were standardized by using the minimum value, the lower quartile (25%), and upper quartile (75%), and the maximum value of each metric dataset [44]. The scoring system of 1, 3, or 5 is representative of the respective threshold value (the 25th percentile and the 75th percentile) of each metric component, according to its response to environmental degradation [37,41]. For the metrics where numbers were expected to decrease with increasing disturbance or pollution, a score of 5 was awarded if the metric value of reference sites was above or equal to the lower quartile (25%), and a score of 3 and 1 was given for the bisected metric value of reference sites with less than the lower quartile. On the other hand, metrics that were expected to increase with increasing disturbance or pollution were awarded a score of 5 if the metric value was equal to or lower than the upper quartile (75%) of the reference sites and a score of 3 and 1 was assigned by the bisected metric value of reference sites with more than the upper quartile [34]. Thus, the appropriate quartile was used as a threshold depending on the type of response to degradation. A score of 5 indicated that the sample was part of the reference groups, a score of 3 indicated an intermediate condition, and a score of 1 indicated the highest deviation from the expected numbers for the reference sites [68].

After aggregating the individual core metric scores into a final MMI value for each sampling site, the index range was then divided into five classes: Class A (Excellent), which

indicated that the site was comparable in condition to the reference biological conditions; Class B (Good) reflecting slight disturbance; Class C (Fair) indicating moderate disturbance; Class D (Poor) indicating highly disturbed biological integrity; and Class E (Impaired) reflecting severely disturbed biological integrity.

The sensitivity of Lao MMI was determined by assessing whether there was clear discrimination between the classified site group, i.e., the reference and stressed sites. The samples collected in 2016–2018 (n = 40) were used to validate Lao MMI for its appropriateness in assessing the water quality of wadeable rivers and streams in Lao PDR. Box-and-whisker plots were performed to distinguish the MMI value of the reference and stressed sites. Spearman's correlation was used to test a correlation between the Lao MMI and some environmental variables.

## 3. Results

### 3.1. Site Classification and Environmental Characteristics

Of the 40 sampling sites from 10 localities, the site classification screening procedure identified seven reference sites (HU-C16, HU-C17, NT-C16, NT-C17, KT-C16, KT-H18, and NN-C16) and seven stressed sites (HL-C16, KN-C16, KH-N17, BL-H17, BL-C17, KH-H18, BL-H18). In the reference sites identified, most of them were mainly distributed in forest areas that were minimally impacted by human activities (Supplementary Table S1). On the contrary, the stressed sites were the areas characterized by high levels of agricultural land uses indicating anthropogenic activities (Supplementary Table S1).

Based on the data of physicochemical parameters and physical and benthic macroinvertebrate characteristics, as shown in Supplementary Table S1, the level of impairment of the studied sites was assessed and categorized into three groups, i.e., reference, intermediate, and stressed sites (Table 2). For statistical analysis, one-way ANOVA was used to test significant differences in the THS (Table 2) and total taxa (Table 3) due to the fact that both data had a normal distribution. While the Kruskal–Wallis test was used to determine significant differences in other data, including environmental variables (Table 2), the number of benthic macroinvertebrate individuals (Table 4), and the number of taxa and individuals in each order and data (Tables 3 and 4) were not normally distributed. The Man-Whitney U test was used to determine statistical differences in the number of taxa and individuals in the order of Isopoda, Orthoptera, Unionoida, and Veneroida (Tables 3 and 4), due to these orders only having datasets and the distribution of data was not normal.

**Table 2.** Environmental variables of reference, intermediate, and stressed sites. Data are expressed as Mean ± SD.

| Parameters | Reference Group (n = 7) | Intermediate Group (n = 26) | Stressed Group (n = 7) | *p*-Value |
|---|---|---|---|---|
| Air temperature (°C) | 24.64 ± 4.53 | 27.12 ± 4.86 | 30.64 ± 4.19 | 0.152 |
| Water temperature (°C) | 22.59 ± 2.16 | 24.73 ± 2.69 | 25.91 ± 3.50 | 0.103 |
| Water channel width (m) | 19.85 ± 11.94 | 17.62 ± 10.20 | 14.18 ± 11.04 | 0.592 |
| Water depth (cm) | 21.35 ± 7.31 | 30.82 ± 13.99 | 33.43 ± 14.02 | 0.182 |
| Water velocity (m/s) | 0.49 ± 0.18 | 0.43 ± 0.18 | 0.32 ± 0.21 | 0.249 |
| Turbidity (NTU) | 2.83 ± 1.84 | 2.19 ± 1.93 | 2.56 ± 2.43 | 0.772 |
| Suspended solids (mg/L) | 5.38 ± 2.96 | 6.19 ± 3.98 | 6.05 ± 2.16 | 0.743 |
| Electrical conductivity (µS/cm) | 113.33 ± 65.56 [a] | 256.40 ± 124.29 [b] | 379.48 ± 101.02 [b] | 0.001 * |
| Total dissolved solids (mg/L) | 58.31 ± 34.23 [a] | 132.09 ± 64.03 [b] | 194.57 ± 53.45 [b] | 0.001 * |
| Dissolved oxygen (mg/L) | 6.30 ± 0.16 | 6.63 ± 0.75 | 6.62 ± 1.78 | 0.325 |
| pH | 7.74 ± 0.48 | 7.91 ± 0.36 | 7.85 ± 0.46 | 0.665 |
| Nitrate nitrogen (mg/L) | 0.20 ± 0.05 [a] | 0.30 ± 0.12 [b] | 0.32 ± 0.10 [b] | 0.021 * |
| Orthophosphate (mg/L) | 0.17 ± 0.06 | 0.19 ± 0.10 | 0.24 ± 0.13 | 0.647 |
| Biochemical oxygen demand (mg/L) | 1.45 ± 0.68 | 1.14 ± 0.81 | 1.11 ± 0.84 | 0.573 |
| Chlorophyll *a* (µg/L) | 0.83 ± 0.20 | 0.97 ± 0.43 | 1.15 ± 0.82 | 0.786 |
| THS | 152.57 ± 14.33 [a] | 137.27 ± 17.40 [a] | 80.71 ± 21.78 [b] | 0.000 * |

Within each row of means different letters (a, b) indicate significant differences at *p* < 0.05 according to pairwise comparisons of the Kruskal–Wallis test and LSD multiple comparisons of one-way ANOVA. * Significant difference at *p* < 0.05.

**Table 3.** Numbers of benthic macroinvertebrate taxa found in reference, intermediate, and stressed sites. Data are expressed as Mean ± SD.

| Taxa | Reference Group | Intermediate Group | Stressed Group | *p*-Value |
|---|---|---|---|---|
| ANNELIDA | | | | |
| Oligochaeta | 0.14 ± 0.38 | 0.08 ± 0.27 | 0.29 ± 0.49 | 0.338 |
| ARTHROPODA | | | | |
| Decapoda | 0.43 ± 0.53 | 0.96 ± 0.96 | 1.14 ± 0.69 | 0.247 |
| Isopoda | 0.29 ± 0.49 | 0.04 ± 0.20 | 0 | 0.330 |
| Coleoptera | 8.14 ± 4.98 | 6.85 ± 2.94 | 4.14 ± 1.68 | 0.085 |
| Collembola | 0 | 0.04 ± 0.20 | 0 | - |
| Diptera | 3.71 ± 1.11 | 4.92 ± 2.33 | 4.57 ± 0.98 | 0.395 |
| Ephemeroptera | 14.29 ± 1.80 [a] | 13.15 ± 2.71 [ab] | 10.29 ± 3.20 [b] | 0.038 * |
| Hemiptera | 4.71 ± 2.43 | 4.96 ± 2.22 | 3.00 ± 2.24 | 0.109 |
| Lepidoptera | 0.71 ± 0.76 | 1.15 ± 1.05 | 0.57 ± 0.98 | 0.298 |
| Megaloptera | 0.71 ± 0.49 | 0.50 ± 0.51 | 0.43 ± 0.53 | 0.522 |
| Odonata | 5.14 ± 3.39 | 4.96 ± 2.24 | 4.00 ± 2.31 | 0.593 |
| Orthoptera | 0.29 ± 0.49 | 0.15 ± 0.37 | 0 | 0.620 |
| Plecoptera | 2.14 ± 0.90 [a] | 1.92 ± 1.74 [ab] | 0.71 ± 0.49 [b] | 0.029 * |
| Trichoptera | 10.57 ± 2.94 [a] | 7.85 ± 2.91 [ab] | 4.86 ± 2.48 [b] | 0.006 * |
| MOLLUSCA | | | | |
| Mesogastropoda | 0.57 ± 0.53 | 1.19 ± 0.94 | 2.00 ± 1.41 | 0.072 |
| Unionoida | 0 | 0.04 ± 0.20 | 0.29 ± 0.49 | 0.330 |
| Veneroida | 0 | 0.31 ± 0.47 | 0.71 ± 0.49 | 0.109 |
| Total taxa | 51.86 ± 15.20 [a] | 49.08 ± 7.56 [a] | 37.00 ± 9.00 [b] | 0.009 * |

Within each row of means different letters (a, b) indicate significant differences at *p* < 0.05 according to pairwise comparisons of the Kruskal–Wallis test and LSD multiple comparisons of one-way ANOVA. * Significant difference at *p* < 0.05.

**Table 4.** Relative abundance of benthic macroinvertebrates found in reference, intermediate, and stressed sites. Data are expressed as Mean ± SD.

| Taxa | Reference Group | Intermediate Group | Stressed Group | *p*-Value |
|---|---|---|---|---|
| ANNELIDA | | | | |
| Oligochaeta | 0.29 ± 0.76 | 0.12 ± 0.43 | 1.71 ± 4.11 | 0.320 |
| ARTHROPODA | | | | |
| Decapoda | 0.43 ± 0.53 | 3.00 ± 4.85 | 5.86 ± 8.65 | 0.089 |
| Isopoda | 0.43 ± 0.79 | 0.12 ± 0.59 | 0 | 0.352 |
| Coleoptera | 35.29 ± 35.00 | 34.46 ± 27.87 | 25.00 ± 35.42 | 0.639 |
| Collembola | 0 | 0.04 ± 0.20 | 0 | - |
| Diptera | 32.00 ± 32.34 [a] | 38.50 ± 23.03 [a] | 78.57 ± 61.95 [b] | 0.045 * |
| Ephemeroptera | 126.57 ± 62.66 | 122.92 ± 51.90 | 96.57 ± 48.59 | 0.539 |
| Hemiptera | 19.14 ± 15.70 | 20.73 ± 13.53 | 18.57 ± 17.93 | 0.801 |
| Lepidoptera | 5.00 ± 10.65 | 2.23 ± 2.69 | 1.14 ± 2.04 | 0.489 |
| Megaloptera | 2.00 ± 2.16 | 1.42 ± 2.27 | 0.43 ± 0.53 | 0.311 |
| Odonata | 12.71 ± 8.60 | 17.69 ± 15.29 | 26.43 ± 23.96 | 0.352 |
| Orthoptera | 0.29 ± 0.49 | 0.23 ± 0.65 | 0 | 0.651 |
| Plecoptera | 11.71 ± 11.60 [a] | 13.81 ± 20.54 [a] | 1.00 ± 0.82 [b] | 0.020 * |
| Trichoptera | 84.29 ± 28.15 | 80.15 ± 51.44 | 62.43 ± 70.96 | 0.450 |
| MOLLUSCA | | | | |
| Mesogastropoda | 10.71 ± 25.74 | 7.38 ± 9.74 | 9.43 ± 12.66 | 0.374 |
| Unionoida | 0 | 0.08 ± 0.39 | 1.86 ± 3.29 | 0.308 |
| Veneroida | 0 | 0.73 ± 1.28 | 4.00 ± 4.55 | 0.060 |
| Total individuals | 340.86 ± 19.99 | 343.62 ± 16.11 | 333.00 ± 17.30 | 0.168 |
| Chironomidae | 17.57 ± 21.83 [a] | 20.19 ± 10.68 [b] | 63.14 ± 68.04 [c] | 0.005 * |

Within each row of means different letters (a, b, c) indicate significant differences at *p* < 0.05 according to pairwise comparisons of the Kruskal Wallis test. * Significant difference at *p* < 0.05.

The water of these site groups was moderately oxygenated with a slightly neutral pH. However, the Kruskal–Wallis test revealed that intermediate and stressed site groups showed higher average values of the EC, TDS, and $NO_3$-N than those of the reference sites ($p < 0.05$) (Table 2). For the THS, the value of the reference site was significantly higher

than those of stressed sites, but there was no significant difference in the total habitat score between the reference and the intermediate sites (Table 2).

### 3.2. Structure of Benthic Macroinvertebrate Community

A total of 13,651 benthic macroinvertebrate individuals in the samples were sorted and identified. They were classified into 3 phyla, 17 orders (Table 3), 102 families, and 241 taxa (Supplementary Table S2). Among them, insects (Arthropoda) were the most diverse group, with 220 taxa belonging to 11 orders and 86 families (91.29 %), followed by Mollusca (6.22 %) and Annelida (0.41 %) (Supplementary Table S2). The dominant groups of insects with the highest taxa richness were Ephemeroptera (18.62%), followed by Coleoptera (16.18%), Trichoptera (14.94 %), and Odonata (14.94%). Based on the relative total abundant and occurrence taxa (Supplementary Table S2), the Chironomidae (Diptera) showed the highest percentages (7.985% and 100%), followed by *Choroterpes* (*Dilatognathus*) (7.457%, 75%), *Cheumatopsyche* (5.487%, 87.5%), *Potamyia* (5.260%, 55%), and *Baetis* (3.780%, 85%), respectively. It was noted that the Chironomidae showed higher abundance in the stressed sites (Table 4). Some representative families and genera of these benthic macroinvertebrates are shown in Figure 2.

The highest number of taxa (taxa richness) of benthic macroinvertebrates was found in Nam Thang (128 taxa) and the lowest in Nam Hinboun downstream (77 taxa). The total taxa richness, Ephemeroptera taxa richness, Plecoptera taxa richness, and Trichoptera taxa richness were significantly different between stream conditions ($p < 0.05$). It was observed that these four taxa richness values were higher in reference sites than those of the stressed sites, while intermediate sites had moderate values of the taxa richness between the reference and stressed sites (Table 3). Moreover, the relative abundance of Plecoptera was significantly decreased in the stressed sites, while the relative abundance of Diptera and Chironomids was considerably higher at the stressed sites (Table 4).

The CCA ordination analysis showed the differences in benthic macroinvertebrate fauna between the three sites as explained by THS, water velocity (VEL), water channel width (WD), EC, and TDS. CCA axis 1 was primarily related to the gradient of THS and VEL, while axis 2 was related to EC, TDS, and WD. All sites in the reference sites were positively correlated to high THS and VEL. The reference sites were also clearly separated from the stressed sites, which were correlated to high EC and TDS. Intolerant taxa (tolerance values 1–3), including EPT, were prominently on the positive side of the CCA axis 1, which was correlated to high THS and VEL. The moderately tolerant taxa (tolerance values 7–8), e.g., some Ephemeroptera, Diptera, Odonata, Mollusca, and tolerant taxa (tolerance values 9–10), such as Chironomidae, Culicidae, and Oligochaeta were abundant on the negative side of the CCA axis 1, which was related to high EC and TDS (Figure 3). In addition, the ordination results confirmed the same trend as the previous results and strongly supported the highly significant relationships between physicochemical variables and benthic macroinvertebrate communities ($r = 0.813$, $p = 0.001$).

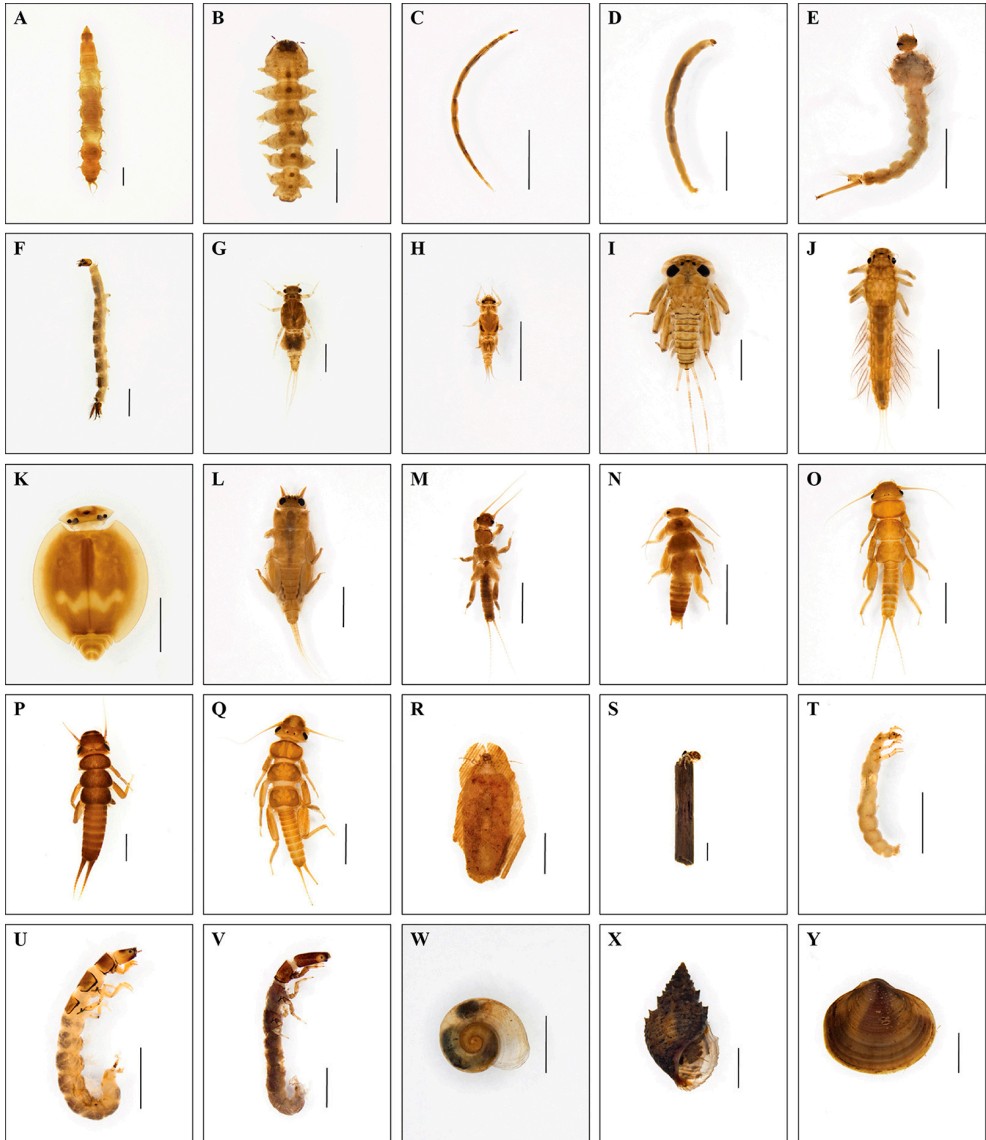

**Figure 2.** Benthic macroinvertebrates found in the study localities with their tolerance values. A-F Diptera: (**A**) Athericidae, *Suragina*, tolerance value three; (**B**) Blephariceridae, *Blepharicera*, tolerance value one; (**C**) Ceratopogonidae, *Bezzia*, tolerance value seven; (**D**) Chironomidae, tolerance value nine; (**E**) Culicidae, tolerance value nine; (**F**) Dixidae, tolerance value three: (**G–L**) Ephemeroptera: (**G**) Caenidae, *Caenis*, tolerance value seven; (**H**) Caenidae, *Clypeocaenis*, tolerance value seven; (**I**) Heptageniidae, *Thalerosphyrus lamuriensis*, tolerance value two; (**J**) Potamanthidae, *Potamanthus formosus*, tolerance value two; (**K**) Prosopistomatidae, *Prosopistoma*, tolerance value three; (**L**) Vietnamellidae, *Vietnamella thani*, tolerance value one: (**M–Q**) Plecoptera: (**M**) Nemouridae, *Indonemoura*, tolerance value one; (**N**) Peltoperlidae, *Cryptoperla*, tolerance value one; (**O**) Perlidae, *Etrocorema*, tolerance value three; (**P**) Perlidae, *Phanoperla*, tolerance value two; (**Q**) Perlidae, *Togoperla*, tolerance value three: (**R–V**) Trichoptera: (**R**) Calamoceratidae, *Anisocentropus*, tolerance value one; (**S**) Calamoceratidae, *Ganonema*, tolerance value one; (**T**) Glossosomatidae, *Agapetus*, tolerance value one; (**U**) Hydropsychidae, *Cheumatopsyche*, tolerance value seven; (**V**) Stenopsychidae, *Stenopsyche*, tolerance value three: (**W–X**) Mesogastropoda: (**W**) Planorbidae, *Indoplanorbis exustus*, tolerance value eight; (**X**) Thiaridae, *Thiara scabra*, tolerance value eight: (**Y**) Veneroida, Corbiculidae, *Corbicula*, tolerance value eight. Scale bars: 2 mm.

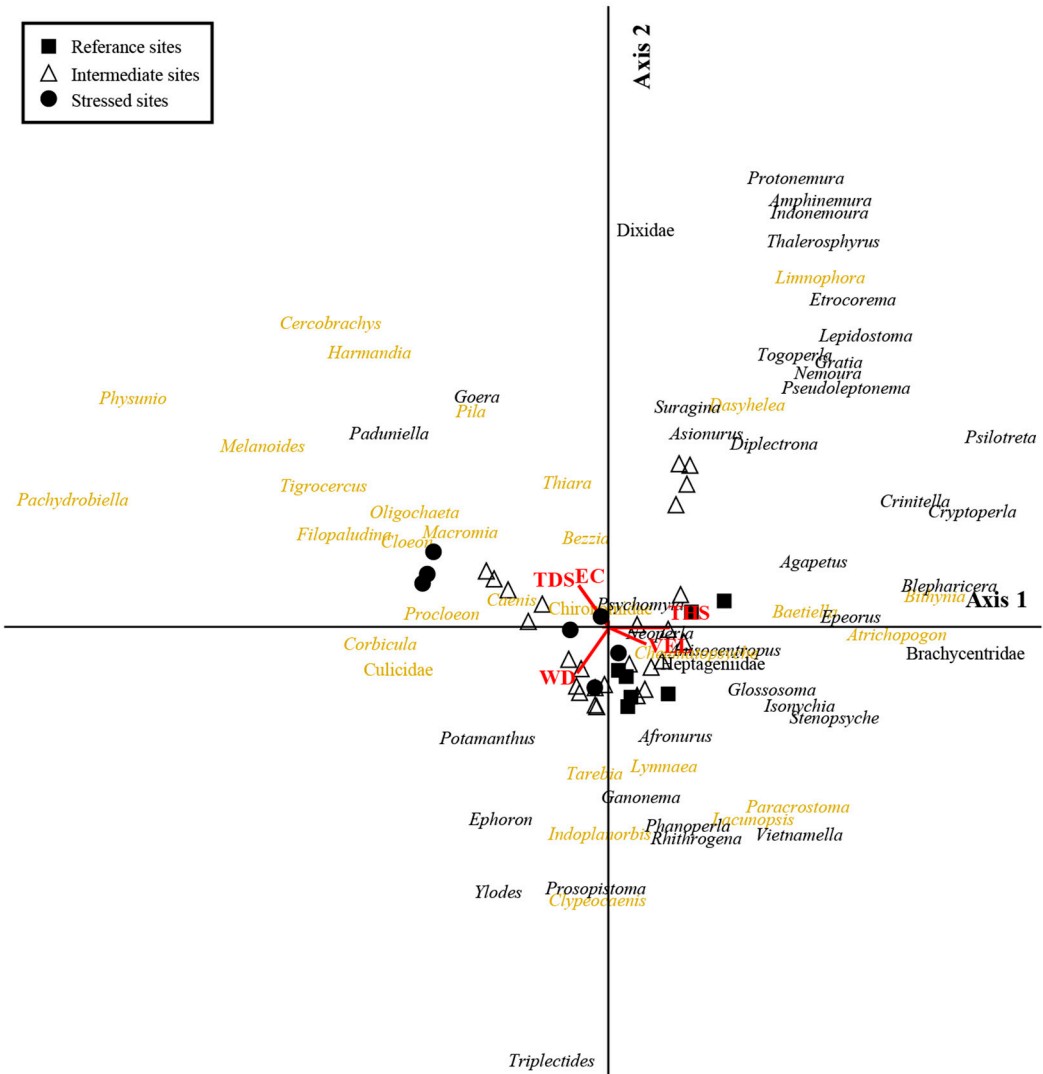

**Figure 3.** CCA ordination analysis showing the association between environmental variables and benthic macroinvertebrate abundance in the reference, intermediate, and stressed sites. Black color denotes intolerant taxa; yellow denotes moderate and tolerant taxa.

### 3.3. Metric Sensitivity and Selection

Before the metric sensitivity test, ANOSIM was used to test the differences in the fauna composition among different ecoregions. The results from this test indicated no significant difference in fauna composition among the three ecoregions ($r$ = 0.1106, $p$ > 0.05). Moreover, an independent sample T-test analysis revealed that the number of benthic macroinvertebrate taxa observed in different seasons (hot and cold) ($p$ = 0.793) and years (2016–2018) ($p$ = 0.651) were not significantly different. Then, box-and-whisker plot analysis was used to test if a metric was sensitive enough, i.e., if it could be used to discriminate between the reference and stressed sites. Of the 54 metrics initially evaluated, 35 metrics were considered sensitive showing a significant difference between the reference and stressed sites (Supplementary Figures S1–S5). The remaining 35 metrics represented five categories of benthic macroinvertebrates, i.e., richness (seven metrics), composition (eight metrics), tolerance (seven metrics), trophic structure (nine metrics), and habit (four metrics). In the sensitivity test, the discrimination power score (sensitivity score), together with a higher DE value (>70%), was used to select potential metrics to integrate into MMI. Metrics with a discriminatory power score of three and the highest DE value were mainly selected as potential metrics. If discriminatory power two was selected, a high (>70%)

discrimination efficiency (DE) was also considered. In addition, a redundancy test was used to select the potential metrics. According to Pearson's correlation analysis, metrics with a high redundancy showed high correlation values ($r > 0.85$) with each other, and those metrics represent how a similar category of the macroinvertebrate community was excluded from the candidate metrics. For example, in the tolerance/intolerance metric, the intolerant taxa metric was redundant with Beck's biotic index metric ($r = 0.863$, $p < 0.01$); thus, the intolerant taxa metric was eliminated. Another example is the richness metric that Trichoptera taxa and EPTC taxa metrics were discarded due to redundancy with the total taxa metric ($r = 0.897$, $0.979$, respectively, $p < 0.05$) (Supplementary Table S3). Apart from sensitivity and redundancy tests, ecological importance and the wider applicability of each metric were also used as criteria for selecting redundant metrics into final core metrics. Moreover, in order to obtain MMI covering five categories of benthic macroinvertebrates, at least one metric per category was retained. Regarding these criteria, a total of 11 core metrics that met the sensitivity and redundancy criteria with characteristics of simplicity, cost-benefit, and common use for biomonitoring were finally selected to integrate into the MMI. The 11-core metrics of the Lao MMI consisted of three richness metrics (Total taxa, EPT taxa, Ephemeroptera taxa), two composition metrics (%Diptera and %Pleoptera), three tolerance metrics (%Tolerant, Beck's Biotic Index, and %Intolerant), one functional feeding group metric (Filterers taxa), and two habit metrics (%Sprawlers and %Burrowers) (Table 5). All of the 11-core metrics clearly distinguished reference and stressed sites (Figure 4). None of them showed partial or considerable interquartile overlaps. A non-parametric analysis utilizing Spearman's rank correlation was conducted to examine the relationship between environmental variables and eleven metrics related to benthic macroinvertebrates. The results, presented in Supplementary Table S4, indicate that a number of the selected metrics exhibited a significant correlation with physicochemical parameters and sediment grain size. Specifically, metrics related to the total taxa (richness category), %Plecoptera (composition category), %Intolerant (tolerant/intolerant category), and Filterer taxa (functional feeding group category) were found to have a negative correlation with physical parameters such as EC and TDS. Additionally, the Filterer taxa and %Burrowers (habit category) metrics were found to have a negative correlation with the chemical parameter, i.e., nitrates. Furthermore, these metrics also demonstrated either a positive or negative correlation with the sediment grain size. For example, the total taxa, %intolerant, and filterer taxa metrics were positively correlated with boulder and pebble, but negatively correlated with sand and muck-mud, with the exception of the total taxa, which had a negative correlation with only muck-mud. Conversely, the %Burrower metric was found to be positively correlated with gravel, sand, and muck-mud but negatively correlated with boulder and cobble.

**Table 5.** Thirty-five candidate metrics selected for the development of MMI and their predicted response to disturbance or pollution, discrimination power score, and DE value. Metrics that met or did not meet the test criteria are expressed as symbols (/) and (-), respectively.

| Metric | Expected Response of Metrics to Pollution | Discrimination Power Score | DE Value | Metric Selection |
|---|---|---|---|---|
| **Richness category** | | | | |
| 1. Total taxa | Decrease | 2 | 71.43 | / |
| 2. EPT taxa | Decrease | 3 | 100.00 | / |
| 3. Ephemeroptera taxa | Decrease | 3 | 71.43 | / |
| 4. Plecoptera taxa | Decrease | 3 | 100.00 | - |
| 5. Trichoptera taxa | Decrease | 3 | 85.71 | Redundant |
| 6. Coleoptera taxa | Decrease | 1 | 42.86 | - |
| 7. EPTC taxa | Decrease | 3 | 100.00 | Redundant |
| **Composition category** | | | | |
| 8. %EPT | Decrease | 1 | 57.14 | - |
| 9. %Ephemeroptera | Decrease | 1 | 42.86 | - |
| 10. Margalef's index | Decrease | 2 | 71.43 | - |
| 11. %Odonata | Increase | 2 | 57.14 | - |
| 12. %Chironomidae | Increase | 2 | 57.14 | - |
| 13. %Diptera | Increase | 3 | 71.43 | / |
| 14. %Plecoptera | Decrease | 3 | 100.00 | / |
| 15. %Trichoptera | Decrease | 2 | 57.14 | - |
| **Tolerance value category** | | | | |
| 16. Intolerant taxa | Decrease | 3 | 100.00 | Redundant |
| 17. %Tolerant | Increase | 3 | 85.71 | / |
| 18. %Dominant taxon | Increase | 3 | 71.43 | - |
| 19. Beck's Biotic Index | Decrease | 3 | 100.00 | / |
| 20. Simpson Index | Increase | 2 | 71.43 | - |
| 21. Hilsenhof's Biotic Index | Increase | 3 | 100.00 | Redundant |
| 22. %Intolerant | Decrease | 3 | 100.00 | / |
| **Functional feeding group category** | | | | |
| 23. %Filterers | Decrease | 1 | 57.14 | - |
| 24. %Scrapers | Decrease | 3 | 71.43 | - |
| 25. %Collectors | Increase | 2 | 57.14 | - |
| 26. Collector taxa | Decrease | 1 | 57.14 | - |
| 27. Filterer taxa | Decrease | 3 | 100.00 | / |
| 28. Predator taxa | Decrease | 2 | 42.86 | - |
| 29. Scraper taxa | Decrease | 3 | 57.14 | - |
| 30. %Shredders | Increase | 1 | 42.86 | - |
| 31. Shredder taxa | Decrease | 3 | 71.43 | - |
| **Habit category** | | | | |
| 32. Clinger taxa | Decrease | 3 | 100.00 | Redundant |
| 33. %Clingers | Decrease | 2 | 71.43 | - |
| 34. %Sprawlers | Increase | 3 | 85.71 | / |
| 35. %Burrowers | Increase | 3 | 85.71 | / |

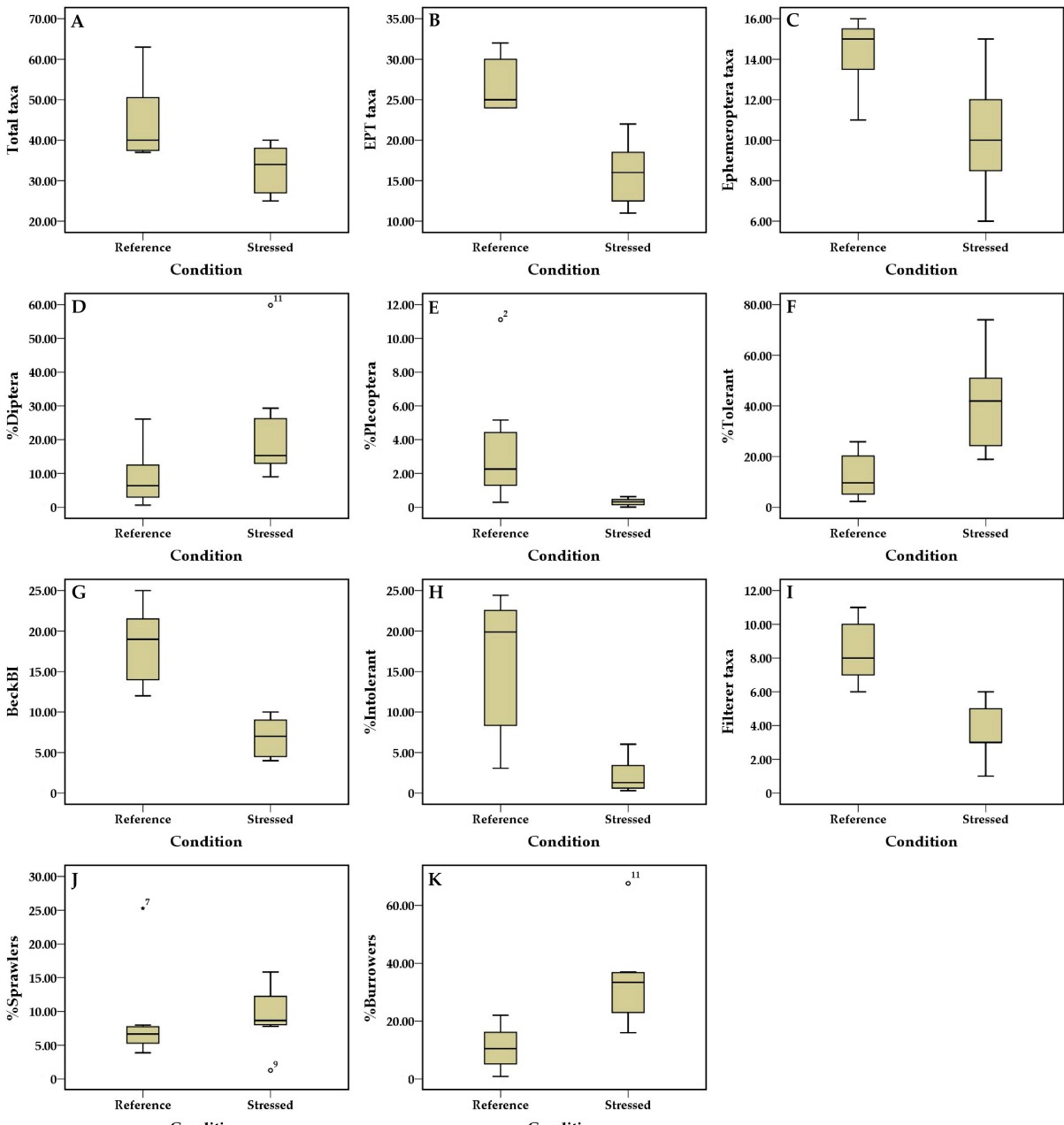

**Figure 4.** Box-and-whisker plots for each of the 11 core metrics sensitive to discrimination between reference and stressed sites for MMI development. (**A**) Total taxa, (**B**) EPT taxa, (**C**) Ephemeroptera taxa, (**D**) %Diptera, (**E**) %Plecoptera, (**F**) %Tolerant, (**G**) BeckBI, (**H**) %Intolerant, (**I**) Filterer taxa, (**J**) %Sprawlers, (**K**) %Burrowers.

### 3.4. Development of Lao MMI

After selecting 11 metrics for integration into the Lao MMI, the value for the appropriate quartile of each of the 11 selected metrics at the reference sites was used as a threshold for segregating the maximum possible score from the lower score (Table 6). Using the metric scores in Table 6, the MMI was calculated by aggregating the scores of each of the 11 core metrics. The range of possible scores for the Lao MMI was determined by the minimum and maximum scores from <13 to ≥53. The range of the Lao MMI was then quadrisected in order to establish the five classes of water quality assessment; excellent condition (score ≥ 53), which pertained to the desired reference biological condition with a low degree of alteration in biological integrity; good (49–52) with good water quality and slightly disturbed biological integrity; fair (25–48) with fair quality of water and moderately

disturbed biological integrity; poor (13–24), with poor quality of water and highly disturbed biological integrity, and impaired condition (score < 13) with very poor water quality and severe impairment of biological integrity (Table 7).

**Table 6.** Score thresholds for each of the 11 core metrics integrated in Lao MMI.

| Metric | Expected Response to Pollution | Statistic Value of Reference Sites | | | | | Scoring Criteria | | |
|---|---|---|---|---|---|---|---|---|---|
| | | Min. | 25th | 50th | 75th | Max. | 5 | 3 | 1 |
| **Richness category** | | | | | | | | | |
| Total taxa | Decrease | 37 | 37 | 40 | 54 | 63 | $\geq$37 | 36–19 | <19 |
| EPT taxa | Decrease | 24 | 24 | 25 | 30 | 32 | $\geq$24 | 23–12 | <12 |
| Ephemeroptera taxa | Decrease | 11 | 13 | 15 | 16 | 16 | $\geq$13 | 12–7 | <7 |
| **Composition category** | | | | | | | | | |
| %Diptera | Increase | 0.6 | 1.9 | 6.3 | 14.4 | 26.1 | $\leq$14.4 | 14.5–21.6 | >21.6 |
| %Plecoptera | Decrease | 0.3 | 0.6 | 2.3 | 5.2 | 11.1 | $\geq$0.6 | 0.5–0.3 | <0.3 |
| **Tolerance value category** | | | | | | | | | |
| %Tolerant | Increase | 2.3 | 4.4 | 9.6 | 25 | 25.8 | $\leq$25 | 25.1–37.5 | >37.5 |
| Beck's Biotic Index | Decrease | 12 | 14 | 19 | 23 | 25 | $\geq$14 | 13–7 | <7 |
| %Intolerant | Decrease | 3.1 | 6 | 19.9 | 23.5 | 24.4 | $\geq$6 | 5.9–3.0 | <3.0 |
| **Functional feeding category** | | | | | | | | | |
| Filterers taxa | Decrease | 6 | 6 | 8 | 11 | 11 | $\geq$6 | 5.0–3.0 | <3 |
| **Habit category** | | | | | | | | | |
| %Sprawlers | Increase | 3.9 | 4.9 | 6.7 | 8 | 25.3 | $\leq$8.0 | 8.1–12.0 | >12.0 |
| %Burrowers | Increase | 0.9 | 1.3 | 10.5 | 19.7 | 22 | $\leq$19.7 | 19.8–28.5 | >28.5 |

**Table 7.** Proposed water quality classes and index score for the studied wadeable rivers and streams in Lao PDR.

| Stream Quality Class | Percentile | Index Score |
|---|---|---|
| Excellent | $\geq$75th | 53 |
| Good | $\geq$25th | 49–52 |
| Fair | <25th | 25–48 |
| Poor | - | 13–24 |
| Impaired | - | <13 |

*3.5. Validation of the Multimetric Index*

In the assessment of the water quality of wadeable rivers and streams in Lao PDR, the developed Lao MMI was tested at the 14 sampling sites (seven reference and seven stressed sites) and was used to develop the index and another 26 sampling sites. Interestingly, the Lao MMI was able to discriminate the reference and stressed site groups (Table 8, Figure 5). Among the forty sampling sites tested, ten (25%) showed waters of excellent quality, four (10%) presented good quality, twenty-four (60%) had fair quality, and two (5%) had poor quality. On the other hand, the conventional physicochemical method established by the Department of Pollution Control of Lao PDR could not clearly distinguish the water quality of 40 sampling sites into different classes: excellent (one site), good (thirty-eight sites) and fair (one site) conditions (Table 8). Therefore, the Lao MMI developed in this study was sensitive enough to distinguish not only the reference and stressed sites but also intermediate environmental conditions. The Lao MMI developed in this study also showed a correlation with some environmental variables using Spearman's correlation analysis. For example, THS was positively correlated with the index (r = 0.610, $p < 0.001$). In contrast, there were negative correlations with EC (r = −0.649, $p < 0.001$), TDS (r = −0.647, $p < 0.001$), and $NO_3$-N (r = −0.348, $p < 0.05$) (Supplementary Table S5). As increasing EC, TDS, and $NO_3$-N values were related to the sites closest to agricultural areas

(Supplementary Table S1), this suggested that the developed MMI had good responsiveness to anthropogenic disturbance.

**Table 8.** Water quality of 40 sampling sites (wadeable rivers and streams) in Lao PDR assessed by using surface water quality standards of Lao PDR (physicochemical method) and the developed Lao MMI.

| Sampling Site | Condition | The Surface Water Quality Standard of Lao PDR | | Lao MMI | |
|---|---|---|---|---|---|
| | | Stream Class | Water Quality | Index Score | Water Quality |
| Cool season 2016 | | | | | |
| HUC16 | Reference | 2 | Good | 49 | Good |
| HLC16 | Stressed | 2 | Good | 33 | Fair |
| KDC16 | Intermediate | 2 | Good | 47 | Fair |
| KNC16 | Stressed | 2 | Good | 31 | Fair |
| NSC16 | Intermediate | 2 | Good | 53 | Excellent |
| NTC16 | Reference | 2 | Good | 55 | Excellent |
| BUC16 | Intermediate | 2 | Good | 45 | Fair |
| BLC16 | Intermediate | 2 | Good | 31 | Fair |
| KTC16 | Reference | 2 | Good | 53 | Excellent |
| NNC16 | Reference | 2 | Good | 53 | Excellent |
| Hot season 2017 | | | | | |
| HUH17 | Intermediate | 2 | Good | 35 | Fair |
| HLH17 | Intermediate | 2 | Good | 39 | Fair |
| KDH17 | Intermediate | 2 | Good | 53 | Excellent |
| KNH17 | Stressed | 2 | Good | 37 | Fair |
| NSH17 | Intermediate | 1 | Excellent | 43 | Fair |
| NTH17 | Intermediate | 2 | Good | 53 | Excellent |
| BUH17 | Intermediate | 2 | Good | 27 | Fair |
| BLH17 | Stressed | 2 | Good | 17 | Poor |
| KTH17 | Intermediate | 2 | Good | 53 | Excellent |
| NNH17 | Intermediate | 2 | Good | 53 | Excellent |
| Cool season 2017 | | | | | |
| HUC17 | Reference | 2 | Good | 43 | Fair |
| HLC17 | Intermediate | 2 | Good | 53 | Excellent |
| KDC17 | Intermediate | 2 | Good | 45 | Fair |
| KNC17 | Intermediate | 2 | Good | 39 | Fair |
| NSC17 | Intermediate | 2 | Good | 45 | Fair |
| NTC17 | Reference | 2 | Good | 51 | Good |
| BUC17 | Intermediate | 2 | Good | 33 | Fair |
| BLC17 | Stressed | 2 | Good | 37 | Fair |
| KTC17 | Intermediate | 2 | Good | 53 | Excellent |
| NNC17 | Intermediate | 2 | Good | 51 | Good |
| Hot season 2018 | | | | | |
| HUH18 | Intermediate | 2 | Good | 39 | Fair |
| HLH18 | Intermediate | 2 | Good | 41 | Fair |
| KDH18 | Intermediate | 2 | Good | 43 | Fair |
| KNH18 | Stressed | 3 | Fair | 35 | Fair |
| NSH18 | Intermediate | 2 | Good | 47 | Fair |
| NTH18 | Intermediate | 2 | Good | 45 | Fair |
| BUH18 | Intermediate | 2 | Good | 47 | Fair |
| BLH18 | Stressed | 2 | Good | 19 | Poor |
| KTH18 | Reference | 2 | Good | 51 | Good |
| NNH18 | Intermediate | 2 | Good | 47 | Fair |

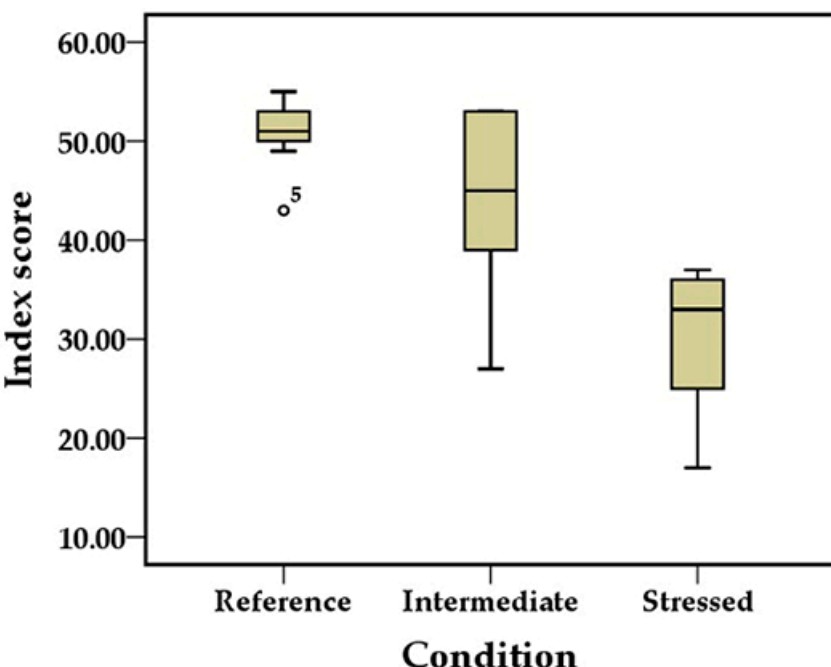

**Figure 5.** Box-and-whisker plot of the final index scores for Lao MMI used to discriminate between reference, intermediate, and stressed sites.

## 4. Discussion

Urbanization, industrialization, and agricultural land use have led to the widespread degradation of water quality in Southeast Asian countries, including the Lao People's Democratic Republic (Lao PDR) [6,8]. The current methods of water quality assessment in Lao PDR primarily focus on physicochemical parameters, with microbial analysis being used as an optional component [69]. The use of benthic macroinvertebrates for biomonitoring water quality, which is commonly used in other regions, has not yet been implemented in Lao PDR. A review of the literature suggests that freshwater biomonitoring in Lao PDR has recently begun through the Mekong River Commission project [47]. Therefore, in this study, a benthic macroinvertebrate-based multimetric index (MMI) was developed to assess the water quality of some wadeable rivers and streams in Lao PDR.

The selection of appropriate reference sites is a fundamental aspect of developing an MMI for biomonitoring and evaluating water quality [37,70]. Ideally, these reference sites should be located in areas that are non-impaired or minimally impaired [70,71]. However, identifying such sites can be challenging, particularly in the lower Mekong region, where many aquatic ecosystems have been disturbed by human activities. To address this issue, the least disturbed areas from anthropogenic activities in each ecoregion were chosen as reference sites in this study. The data on benthic macroinvertebrates, together with the physicochemical properties of waters and habitats from seven reference and seven stressed sites of wadeable rivers and streams, were used to generate the Lao MMI (Supplementary Table S1). A total of 54 candidate metrics from five categories of benthic macroinvertebrates (richness, composition, tolerance/intolerance to pollution, habit, and functional feeding group) were initially considered for integration into the MMI, but some were eliminated due to a lack of effective discrimination between the reference and stressed sites. The final MMI included 11 metrics covering five categories of benthic macroinvertebrates, which were previously used for MMI development in various climatic regions [14,15,17,18,30,32,34,35,37,40–44,72–75] and were selected based on criteria such as cost-benefit, wide applicability, simplicity, and ease of calculation [11,17,19,37].

The taxa richness metric, a measure of the number of different species present, has been shown to be a reliable indicator of anthropogenic disturbance and exhibits good responsiveness to changes in water quality [15,37]. In this study, the total taxa (number of

taxa), which are representative of the richness metric, was sufficiently sensitive to distinguishing reference and stressed sites. Previous research has demonstrated that high species richness is indicative of physical habitat diversity, good water quality, high availability of food resources, undisturbed conditions, and overall ecosystem health [14,15,37]. Furthermore, studies have shown that taxa richness, particularly that of sensitive taxa, decreases substantially in areas impacted by pollution, as pollution-intolerant species are lost [76–78]. In the current study, a decrease in taxa richness was observed in stressed sites, such as the Nam Khan, which may be attributed to moderate increases in nutrient availability from nearby agricultural activities. Previous studies have also reported the use of total taxa, the number of EPT taxa, and the number of Ephemoroptera taxa metrics in developing MMIs [37,41–44,71,79]. These metrics have been found to be excellent indicators in river and stream biomonitoring due to their sensitivity to perturbation and the fact that crucial environmental factors affect their spatial distribution [37].

The EPT (Ephemeroptera, Plecoptera, and Trichoptera) richness metric, which comprises sensitive taxa, is widely acknowledged as a significant bioindicator for determining the water quality of aquatic ecosystems [11,76,80]. Studies have demonstrated that human activities near rivers can have negative effects on the abundance and diversity of EPT taxa [80]. Since they like clean, cold running water, Plecoptera (stoneflies) are among the EPT that is particularly sensitive to environmental changes, specifically changes in water DO and temperature [81,82]. Due to their limited ability to move, stoneflies will inevitably respond to changes in the quality of the water [80,82]. Therefore, the existence of stoneflies suggests that environmental conditions in habitats are within their tolerance range [80]. Ephemeroptera, or mayflies, are another significant component of the EPT and are typically prevalent in highland streams with rather high DO concentrations. With rising habitat disturbance, pollution, and conductive water, the abundance of mayflies drastically decreases [83,84]. Thus, Ephemeroptera can be potentially used as a bioindicator of anthropogenic disturbance, except for some groups, e.g., Baetidae and Caenidae, which are capable of tolerating a broad range of anthropogenic disturbances [19,37,83]. Trichoptera (caddisflies), the final member of the EPT group, and some cased caddisflies construct their cases from leaves and other debris found in rivers and streams [85]. The habitats that are close to forests are likely to obtain available leaf materials from the surrounding trees; hence Trichopterans can use the leaf litter for their case construction and as a food source because they are shredders and collector-gatherers [85,86]. Given the ecological characteristics and functions of the EPT group, the EPT richness metric has become widely used in many biomonitoring programs due to its sensitivity to anthropogenic disturbance [19,80,83].

The composition metric is an essential core metric commonly used in conjunction with the richness metric for MMI development. The high degree of composition and the richness of certain groups of benthic macroinvertebrates, such as EPT taxa, can reflect minimally disturbed areas that provide heterogeneous habitats for the diverse niche partitioning of macroinvertebrate communities and support habitats for a diverse array of macroinvertebrates [86]. In contrast, rivers and streams that have been disturbed due to anthropogenic activities, such as agricultural land use and urbanization, tend to have a low composition, and richness of sensitive taxa, such as EPT [19,80,83]. Given that they are highly sensitive to human disturbance, composition and richness metrics are two core essential components for developing MMIs that have been previously used in the biomonitoring of river and stream ecosystems [11,15,37,41,86]. Additionally, certain macroinvertebrate groups can be tolerant (e.g., Diptera) or intolerant (e.g., EPT group) to pollution and human disturbance [15,37,86]. Diptera, including its diverse Chironomidae family and other tolerant taxa such as Oligochaetes (Annelida) and mollusk Bithynidae, are frequently predominant in freshwater ecosystems that are rich in increasing nutrient concentration and depleting dissolved oxygen concentration [19,52,86,87]. Due to the possession of hemoglobin in their body, Chironomidae and Oligochaetes are able to tolerate and survive in disturbed areas with depleted oxygen concentration because oxygen molecules can be trapped by hemoglobin molecules [52,86,88,89]. Both groups of macroinvertebrates are, therefore,

useful indicators for assessing organic pollution in river and stream ecosystems and are important for the development of ecological integrity MMI and other biomonitoring tools globally [19,86,89].

In this study, two habit metrics, %Burrowers, and %Sprawlers, were selected for incorporation into the established MMI. Habit metrics take into account the mode of locomotion and position of benthic macroinvertebrates in aquatic environments [55]. The percentage of burrowers, primarily composed of chironomid larvae (Diptera), was found to be positively correlated with EC, TDS, nitrate, gravel, and muck mud. Similarly, the percentage of sprawlers, primarily composed of caenid mayfly nymphs, was found to be positively correlated with nitrate, gravel, and sand. Furthermore, EC, TDS, and nitrate values were found to be significantly higher in stressed sites, where the substrate grain size was dominated by fine sediment. This change in the substrate in stressed sites, may be caused by agricultural activities, leading to the deposition of fine sediment due to the instability of bank sides. Previous research also suggests that the tight packing of sand grains and muck mud reduces the trapping of detritus, thus limiting the availability of oxygen concentrations [90]. Thus, the occurrence of fine sediments due to agricultural activity becomes a preferred substratum for dipteran larvae. Similarly, many caenids are able to inhabit the surface of very fine (silty) substrata, with nymphs having the first abdominal gill modified to operculate gill cover and protect the underlying gills from siltation [55].

Regardless of their taxonomic affiliations, many species that obtain food in comparable ways were categorized in the same functional feeding category [91]. In the current study, the functional feeding group metric was included in the MMI as it provided insight into the degradation of ecosystem functioning by measuring the relative abundance and proportion of different functional groups in response to disturbances that affect food availability. It has been found that certain functional feeding groups, such as shredders and scrapers, may be more sensitive to environmental changes, while others, such as gatherers and filterers, may be more tolerant of pollution [34,92]. Many of the filterers found in this study belonged to the family Hydropsychidae (Trichoptera) (Figure 2), which are able to tolerate moderately polluted water [52]. Although Hydropsychidae members are moderately tolerant, they tended to decrease in stressed sites in this study, as previously observed in stream biomonitoring studies in the United States [93] and Thailand [94]. It is important to note that there are limitations to the use of functional feeding groups as it can be difficult to properly assign some taxa to functional feeding groups as species can change their feeding mode during the developmental stage. However, the use of filterer taxa in this study is reliable as most of them are trichopterans in the Hydropsychidae family, which have morpho-behavioral characteristics that are easily classified as filterers [95]. To ensure the proper assignment of functional feeding groups, guidelines from various works in the literature for categorizing the group based on life stage are recommended [15,95].

The assessment of water quality based solely on physicochemical parameters established by the Department of Pollution Control of Lao PDR did not effectively distinguish between the reference and stressed sites. However, the Lao MMI, which is based on benthic macroinvertebrate data, was able to differentiate between reference and stressed sites with a high level of discrimination (Table 8), indicating its sensitivity to environmental changes. Additionally, eleven metrics from the MMI were found to be significantly correlated with environmental variables and sediment grain size (see Supplementary Table S4), suggesting that the MMI was responsive to anthropogenic disturbances that could alter the distribution and abundance of benthic macroinvertebrates in rivers and streams. This is supported by previous research, which has shown that changes in the substrate condition due to anthropogenic and agricultural activity can affect the distribution and abundance of benthic macroinvertebrates [29,96]. For instance, the richness and density of benthic macroinvertebrates clearly decreased in sandy habitats, leading to a reduction in food sources for fish [26].

Prior to being employed in biomonitoring, the development of MMIs typically needs to undergo a stability test for an appropriate amount of time in order to prevent errors or misinterpretations about the health of river and stream ecosystems caused by the seasonal variability of the index [15,17,41]. In some cases, studies have revealed that monsoonal rains can cause seasonal differences in the assemblage structure of macroinvertebrates [97], but other studies have found that seasonality did not affect the ecological status by using MMI in tropical Southeast countries such as Vietnam [17], Thailand [44], and Malaysia [41]. The results of this study are consistent with these reports, and the 11-core metrics are, therefore, sufficiently qualified as biomonitoring components for assessing water quality and ecological integrity in wadeable rivers and streams in Lao PDR.

Rapid biomonitoring methods, which use coarse taxonomy to save time and money, are widely accepted for their practicality [78]. It is acknowledged that the process of identifying benthic organisms at a detailed level is a costly and time-consuming endeavor, requiring a comprehensive library of keys, reference collections, specialized knowledge, techniques, and equipment [78,98]. For detailed identification at the species level, taxonomic knowledge may be incomplete as keys and descriptions of specific life stages or sexes for many species are not readily available or easily accessible. Additionally, the use of morphology-based taxonomies may be limited by intraspecific variability and the difficulty of distinguishing between closely related species [78]. Therefore, the use of a higher taxonomic resolution leads to more accurate and informative results with low cost [78], although the use of a genus or species level could be useful to detect specific pollution [11]. In this study, the Lao MMI was designed for use in a biomonitoring program in the Mekong region, including the Lao PDR, where available keys for the identification of benthic macroinvertebrates at the phylum, class, order, family, and genus levels were established [51]. The advantage of the developed Lao MMI is that it is suitable for ecologists who are not familiar with benthic macroinvertebrate taxonomy, and it saves time and cost as the Lao MMI utilizes coarse taxonomy for sample diagnosis. However, it should be noted that the MMI developed here has some limitations. For example, it can only be used to monitor freshwater ecosystems in the explored ecoregions, particularly the Lower Lancang, Khorat Plateau, and Kratie-Stung Treng, as MMIs developed from different ecoregions or environmental gradients in specific geographical regions must be used with caution due to the differences in reference conditions, anthropogenic pressures, and regional species assemblages [19,37]. Additionally, the application of MMI in various field areas is recommended to ensure the stability of the MMI in the biomonitoring program.

## 5. Conclusions

In the current study, benthic macroinvertebrate-based MMI has been developed to assess the water quality of streams and wadeable rivers in Lao PDR. A total of 40 samples were collected across the country, and 11 core metrics, including Total taxa, EPT taxa, Ephemeroptera taxa, % Diptera, %Plecoptera, %Tolerant, Beck's biotic index, %Intolerant, Filterers taxa, %Sprawlers, and %Burrowers, were selected for incorporation into the final index. The developed Lao MMI was found to be sensitive enough to differentiate between minimal human disturbance (reference sites) and areas of anthropogenic impact (stressed sites). Additionally, the Lao MMI had an advantage over conventional physicochemical methods as it was able to clearly classify the water quality of 40 sampling sites into four classes, while the physicochemical method was unable to do so. Furthermore, the Lao MMI is cost-effective and easy to use for biomonitoring as it primarily utilizes family and EPT genera levels for benthic macroinvertebrate diagnosis. To the best of our knowledge, this research represents the first attempt to utilize a biomonitoring method based on multimeric data generated from benthic macroinvertebrates for monitoring the water quality of rivers and streams in Lao PDR. Further research in other areas of rivers and streams is needed to validate the stability of the Lao MMI in biomonitoring programs.

**Supplementary Materials:** The following supporting information can be downloaded at: https://www.mdpi.com/article/10.3390/w15040625/s1, Table S1: Site classification based on environmental parameter and benthic macroinvertebrate da-ta; Table S2: Data on relative abundance, occurrence, tolerance values (TolVal), functional feeding groups (FFGs), and habits of benthic macroinvertebrates observed from 40 sampling sites; Table S3: Pearson's correlation analysis between candidate metrics of benthic macoinvertebrates in the reference sites; Table S4: Spearman's correlation coefficients among the eleven core metrics, envi-ronmental variables and grian sizes; Table S5: Spearman's correlation between final index scores of Lao MMI and some environmental variables; Figure S1: Box and whisker plots of benthic ma-croinvertebrate metrics to discriminate between reference and stressed sites. A-G richness catego-ry: A, total taxa; B, EPT taxa; C, Ephemeroptera taxa; D, Plecoptera taxa; E, Trichoptera taxa; F, EPTC taxa; G, Coleoptera taxa: H-O composition category: H, %EPT; Figure S2: Box and whisker plots of benthic macroinvertebrate metrics to discriminate between reference and stressed sites. H-O composition category: I, %Ephemeroptera; J, Margalef's index; K, %Odonata; L, %Chirono-midae; M, %Diptera; N, %Plecoptera; O, %Trichoptera: P-V tolerance/intolerance category: P, In-tolerant taxa; Figure S3: Box and whisker plots of benthic macroinvertebrate metrics to discrimi-nate between reference and stressed sites. P-V tolerance/intolerance category: Q, %Tolerant; R, %Dominant taxon; S, Beck's Biotic Index; T, Simpson Index; U, Hilsenhof's Biotic Index; V, %In-tolerant: W-AE functional feeding group category: W, %Filterers; X, %Scrapers; Figure S4: Box and whisker plots of benthic macroinvertebrate metrics to discriminate between reference and stressed sites. W-AE functional feeding group category: Y, %Collectors; Z, Collector taxa; AA, Fil-terer taxa; AB, Predator taxa; AC, Scraper taxa; AD, %Shredders; AE, Shredder taxa: AF-AI habit category: AF, Clinger taxa; Figure S5: Box and whisker plots of benthic macroinvertebrate metrics to discriminate between reference and stressed sites. AF-AI habit category: AG, %Clingers; AH, %Sprawlers; AI, %Burrowers.

**Author Contributions:** Conceptualization, N.S.; methodology, C.H., C.V., J.S., K.R., N.S. and V.V.; validation, J.S., N.S. and W.M.; formal analysis, J.S. and N.S.; investigation, C.H., C.V., J.S., N.S., V.V. and W.M.; resources, N.S.; data curation, J.S. and N.S.; writing—original draft preparation, J.S., N.S. and W.M.; writing—review and editing, C.H., C.V., K.R., N.S., V.V. and W.M.; visualization, J.S., N.S. and W.M.; supervision, N.S.; funding acquisition, N.S. All authors have read and agreed to the published version of the manuscript.

**Funding:** This research was funded by the Center of Excellence on Biodiversity, Thailand, grant number BDC-PG1-159008.

**Data Availability Statement:** Not applicable.

**Acknowledgments:** The authors would like to thank the Department of Natural Resources and Environment of Luangprabang, Xaignabouly, Vientiane, Khammouan and Xekong Provinces for allowing us to access the studied areas.

**Conflicts of Interest:** The authors declare no conflict of interest.

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
