# Peer review of "Benthic Macroinvertebrate Communities in Wadeable Rivers and Streams of Lao PDR as a Useful Tool for Biomonitoring Water Quality: A Multimetric Index Approach"

_water, doi:10.3390/w15040625_

Round 1

Reviewer 1 Report

The authors in the MS was tried to develop and use  what they called 'A Multimetric Index Approach' ecosystem, monitoring. To me its a good approach for the locality/ region.  However, the MS needs some improvements:

1. Abstract: please elaborate MMI in the abstract in first appearance. Mention how it can be used.

2. Introduction: Add a paragraph explaining how benthos are controlled by the environmental  factors. Sediment grain size  is one of the major factor but not included. 

3. Results: As mentioned earlier, please try to add grain size data   in the analyses. I did not see diversity indices - commonly used for benthic data.  Have you used ANOVA for comparing the sites - parametric or non parametric? Was the data distribution checked before ANOVA. Benthic data are are generally non-normal. 

Discussion: Please add the limitation of MMI approach 

Thank you

Author Response

Dear sir,

Thank you for your valuable comments and suggestion that allowed us to greatly improve the quality of the manuscript. We have revised the manuscript as the reviewer suggestion. Please find an attached file response to reviewer.

sincerely yours,

Narumon Sangpradub

Reviewer 2 Report

Overall the manuscript entitled Benthic Macroinvertebrate Communities in Wadeable Rivers and Streams of Lao PDR as a Useful Tool for Biomonitoring of Water Quality: A Multimetric Index Approach is a good multidisciplinary approach to analyse the health of the aquatic ecosystems in LAO PDR, however the manuscript is way to long (honestly I never saw a manuscript like this before) also es very redundant in several sections. Information must be reduced and more specific.

Sometimes statistics are mixed among sections, this is a very important issue to be covered

Those eleven traits you conclude they met the criteria that were incorporated to your final index were cost effective? which is the main objective of biomonitoring is barely discussed and it supossed to be the most fundamental results of the research isnt it?

Other comments are included in the PDF

Author Response

(The authors gave the same response as above.)

Reviewer 3 Report

In this MS, the authors use benthic macroinvertebrate communities to monitor the water quality in Wadeable Rivers and Streams of Lao PDRMMI. In general, MMI approach used in this study is very interesting, which provides a new insight in the area of environment pollution monitoring. I can recommend this MS for publication in WATER after the following concerns are solved. My comments are as follows:

Major concerns:

1.      INTRODUCTION:

1.1   Line 45-48, If possible, I suggest to replace the sentence of “Over 75% of…for rural communities of Lao PDR” by some other data, eg objective pollution data. Cause the description from the authors may cause ethnic discrimination, though the authors describe the facts.

1.2   In this part, the authors detailedly describe MMI in paragraph 2-4. In my opinion, some of these sentences seem a bit redundancy. The authors should simplify this part and move some descriptions to DISCUSSION. In addition, Line 99, the definition of MMI should be moved forward.

2.    RESULTS 2.7 Data Analysis, the authors declare that One-way ANOVA or Kruskal-Wallis test were used to determine significant differences in environmental parameters, please indicate specific statistical method for these 16 parameters in Table 2. Also, in Table 3 and 4.

3.      CONCLUSION should be simplified. Conclusion in a paper usually is used to display the main finding and significance of the study. The authors put too many words in this part, which has already displayed in DISCUSSION.

4.      Figure 4-5 should be re-modified to make them much clearer. Also, their background should be colorless or transparent.

Minor concerns:

1.      Line 19, 21 and 95, abbreviation (MMI) should be indexed when first appeared. Besides, there are many other abbreviations misused, eg Line 15 and 38, Lao PDR. The authors should carefully correct them.

2.      Table 2, 0C ???

3.      Table 2-4, what’s the meaning of superscript letter “a” or “b” superscript? Please clarify the situation.

Author Response

(The authors gave the same response as above.)

Round 2

Reviewer 1 Report

The authors has improved the MS as per my suggestions. 

Author Response

Dear Sir,

Thank you for your comment. The revised manuscript has undergone proofreading and editing by BOSS TRAINING ACADEMY PTY LID, Melbourne, Australia.

Kind Regards,

Narumon Sangpradub

Reviewer 2 Report

Manuscritp improve substancially, so far, there are very little changes that needs to be done: 

Line 345: What kind of pos hoc test did you performed for this analysis?

how do you explain the orthoptera captures? 

figure 4: each comparison should have an asterisk to indicate significant differences, otherwise a NS. Remember that graphs should be selfdescriptive please

Line 593: are this differences significantly different?? please clarify otherwise add a NS sign at the graph

Author Response

Dear Sir,

Thank you for your comment. The revised manuscript has undergone proofreading and editing by BOSS TRAINING ACADEMY PTY LID, Melbourne, Australia. Please find a file of response to reviewer.

Kind Regards,

Narumon Sangpradub
